

# Unshielded Precipitation Gauge Collection Efficiency with Wind Speed and Hydrometeor Fall Velocity. Part II: Experimental Results

Jeffery Hoover[1], Michael E. Earle[1], Paul I Joe[1]

[1]Environment and Climate Change Canada, Toronto, ON, M3H 5T4, Canada

*Correspondence to*: Jeffery Hoover (jeffery.hoover@canada.ca)

**Abstract.** Five collection efficiency transfer functions for unshielded precipitation gauges are presented that compensate for wind-induced collection loss. Three of the transfer functions presented are dependent on wind speed and precipitation fall velocity, and were derived through computational fluid dynamics modelling in Part 1 (CFD function) and from measurement data (HE1 function with fall velocity threshold and HE2 function with linear fall velocity dependence). These functions are

evaluated alongside universal ($K_{Universal}$) and site-specific ($K_{CARE}$) transfer functions with wind speed and temperature dependence. Their performance was assessed using 30-minute precipitation event accumulations reported by unshielded and shielded Geonor T-200B3 precipitation gauges over two winter seasons. The latter gauge was installed in a Double Fence Automated Reference (DFAR) configuration comprising a single-Alter shield within an octagonal, wooden double fence. Estimates of fall velocity were provided by a Precipitation Occurrence Sensor System (POSS).

The CFD function reduced the RMSE (0.08 mm) relative to $K_{Universal}$, $K_{CARE}$, and the unadjusted measurements, with a bias error of 0.011 mm. The HE1 function provided a RMSE of 0.09 mm and bias error of 0.006 mm, capturing well the collection efficiency trends for rain and snow. The HE2 function better captured the overall collection efficiency, including mixed precipitation, resulting in a RMSE of 0.07 mm and bias error of 0.006 mm. The improved agreement demonstrates the importance of fall velocity for collection efficiency.





## 1 Introduction

Automated catchment-type precipitation gauge measurements are critical as references for, and input to, weather, climate, hydrology, transportation, and remote sensing applications. The systematic bias of these gauges due to wind-induced
undercatch is a major challenge, particularly with respect to the measurement of solid precipitation (Rasmussen et al., 2012;Kochendorfer et al., 2018).

Intercomparisons to assess gauge undercatch have demonstrated that an unshielded weighing precipitation gauge can capture less than 50% of the actual amount of solid precipitation falling in air when the wind speed exceeds 5 m s$^{-1}$ (Kochendorfer et al., 2017b). Various adjustment functions have been proposed to compensate for undercatch, based on measured wind speed
and air temperature (Goodison, 1978;Yang et al., 1998;2005;Yang and Simonenko, 2013;Yang, 2014;Rasmussen et al., 2001;Smith, 2007;Wolff et al., 2015;Kochendorfer et al., 2017a).

In the 1998 World Meteorological Organization (WMO) Solid Precipitation Measurement Intercomparison, adjustments were determined experimentally, primarily for manual gauges, by comparison to manual reference measurements of precipitation using a collector with a Tretyakov shield in the WMO Double Fence Intercomparison Reference (DFIR) configuration
(Goodison et al., 1998). Precipitation events were monitored by observers, who reported the amount and type of snow, wind speed, and temperature statistics for each event. Events were defined based on the duration of continuous snowfall when the reference DFIR precipitation accumulation was greater than or equal to 3 mm.

Based on the work of Goodison (1978), Goodison et al. (1998), and Yang et al. (1998), adjustment functions for unshielded gauge collection efficiencies were recommended for snow, mixed precipitation, and rain, based on the wind speed at gauge
height. While these adjustments provided improvements for manual precipitation accumulation measurements, their application to automated measurements at shorter time scales, and where the precipitation type may not be well defined, presents a significant challenge (Colli, 2014;Colli et al., 2014;Colli et al., 2016a;Colli et al., 2016b;Thériault et al., 2015;Thériault et al., 2012).

The WMO commissioned another intercomparison, the Solid Precipitation Intercomparison Experiment (SPICE), to assess
various automated technologies for the measurement of precipitation accumulation and snow depth and to recommend automated field reference systems (Nitu et al., 2018). An automated precipitation gauge configured with a single-Alter shield within a DFIR fence was chosen as the field reference configuration for precipitation accumulation; this was referred to as the Double Fence Automated Reference (DFAR) configuration. For assessment purposes, precipitation events were defined as 30-minute periods with ≥ 0.25 mm precipitation captured by the reference gauge and ≥ 60% (18 minutes) precipitation occurrence
as indicated by a disdrometer. This approach was selected to ensure confidence in reference measurement accumulations relative to gauge uncertainties (e.g. due to wind and temperature), with sufficiently short duration to capture the event conditions (e.g. wind, temperature, precipitation characteristics) within dynamic environments (Kochendorfer et al., 2017a). Transfer functions for unshielded and shielded gauges were derived as an exponential function of wind speed using 30-minute precipitation events from the SPICE data set (Kochendorfer et al., 2017a). Separate functions were developed for solid





precipitation and mixed precipitation defined by air temperature ranges: less than -2 °C for solid precipitation, and between -2 °C and 2 °C for mixed precipitation. These functions were based on an exponential model for collection efficiency as a function of wind speed only (Goodison, 1978).

Using Bayesian analysis of Norwegian measurement data, Wolff et al. (2015) developed a precipitation phase-independent, continuous transfer function with respect to wind speed and air temperature for a single-Alter shielded Geonor precipitation

gauge. A similar, but less complex, function was developed by Kochendorfer et al. (2017a;2018) using the SPICE data set, including results from eight measurement sites in Canada, Norway, Finland, Switzerland, and the USA. The application to precipitation accumulation measurements from unshielded weighing gauges in SPICE was shown to reduce the overall bias relative to the DFAR; however reduction in the root mean square error (RMSE) was less significant (Kochendorfer et al., 2017a;2017b;2018;Wolff et al., 2015). The RMSE values for adjusted measurements were on the order of 0.20 mm

(Kochendorfer et al., 2017a); however, a separate comparison of replicate configurations of weighing gauges with single-Alter, double-Alter, and US small DFIR shields at the US WMO-SPICE site exhibited errors of 0.09 mm, 0.08 mm, and 0.07 mm, respectively (Kochendorfer et al., 2017b), indicating that further reductions in RMSE are possible.

The errors, based on universal adjustments with wind speed and air temperature, can vary significantly by site, presumably driven by differences in climatology (Smith et al., 2020;Kochendorfer et al., 2017a). This has motivated further work on site-

and climate-specific transfer functions (Koltzow et al., 2020;Smith et al., 2020). Another potential avenue for reducing errors in adjusted measurements is by improving the ability of transfer functions to distinguish among different precipitation types and their aerodynamic properties (Thériault et al., 2012;Wolff et al., 2015;Nešpor and Sevruk, 1999).

Computational fluid dynamics (CFD) studies simulate the airflow around precipitation gauges and the associated collection efficiencies for rain and solid precipitation (Nešpor and Sevruk, 1999;Constantinescu et al., 2007;Colli, 2014;Colli et al.,

2014;Colli et al., 2015;Colli et al., 2016a;Colli et al., 2016b;Thériault et al., 2012;Thériault et al., 2015;Baghapour and Sullivan, 2017;Baghapour et al., 2017). These studies have demonstrated the influence of wind speed, turbulence, hydrometeor characteristics (size, density, drag, terminal velocity), and gauge and shield geometry on precipitation gauge undercatch. For rainfall, Nešpor and Sevruk (1999) showed increases in wind-induced error for smaller drop sizes with lower terminal velocities, with errors increasing for higher wind speeds. The conversion factor (inverse of overall collection efficiency) varied

with the precipitation intensity and rainfall type, which influenced the distribution of hydrometeor sizes and terminal velocities. Thériault et al. (2012) demonstrated similar trends for snowfall, with collection efficiencies varying significantly with hydrometeor type and size distribution. Simulated collection efficiencies for wet snow and dry snow hydrometeors captured the general upper and lower bounds of experimental observations, respectively, with the lower collection efficiency for dry snow hydrometeors attributed to their lower terminal velocity and interaction with the local airflow around the gauge.

For a Geonor gauge with single-Alter shield, Thériault et al. (2012) used a constant drag coefficient hydrometeor tracking model to develop a series of adjustments based on wind speed for different hydrometeor types. Colli et al. (2015) extended this work to show the influence of different hydrometeor drag models on collection efficiency results. Empirical drag model results (Khvorostyanov and Curry, 2005), based on the relative hydrometeor-to-air velocity over the hydrometeor trajectory,





were shown to yield higher collection efficiencies compared with constant drag coefficient results that can overestimate drag
values. Colli et al. (2015) developed adjustments based on wind speed for unshielded and single-Alter-shielded gauges for
three specific hydrometeor size distributions. Further studies, using computationally intensive Large Eddy Simulation models,
have better resolved the intensity and spatial extent of turbulence around the gauge orifice, which can lead to temporal
variations in collection efficiency results (Colli et al., 2016a;Colli et al., 2016b;Baghapour and Sullivan, 2017;Baghapour et
al., 2017). The degree of turbulence varies depending on the shielding and wind speed (Baghapour et al., 2017).
Collection efficiency results have been shown to be highly dependent on the hydrometeor fall velocity (Nešpor and Sevruk,
1999;Thériault et al., 2012;Colli et al., 2016b;Hoover et al., 2020). The computational fluid dynamics analysis presented in
Hoover et al. (2020, hereafter Part I) characterized the collection efficiency dependence on wind speed and precipitation fall
velocity for different precipitation types. Collection efficiencies were shown to be similar for different hydrometeor types with
identical fall velocities, enabling the development of a universal transfer function based on wind speed and hydrometeor fall
velocity. The use of precipitation fall velocity offers a physically-based approach to improve adjustment functions by
exploiting the aerodynamic properties of falling precipitation that influence collection efficiency. This information may be
provided by instruments such as present weather sensors and disdrometers.

In this work, adjustment functions are developed and evaluated for unshielded Geonor T-200B3 weighing precipitation gauges.
The unshielded gauge configuration allows for the assessment of a broader range of collection efficiencies, as the degree of
undercatch is generally more pronounced for unshielded gauges relative to shielded configurations. Further, by focussing on
the unshielded configuration, no assumptions are required regarding the behaviour of the shield slats and their role in
momentum reduction and turbulence generation around the gauge. For this study, three transfer functions with wind speed and
fall velocity dependence are assessed, including the CFD transfer function developed in Part I and two other transfer functions
(HE1 and HE2) developed herein. The Precipitation Occurrence Sensor System (POSS) was used to estimate fall velocities
and hydrometeor types for the assessment. These transfer functions are assessed against transfer functions with dependence
on wind speed and air temperature, including one of the universal functions developed by Kochedorfer et al. (2017a) and a
site-specific function determined herein using similar methodology.

## 2 Method

### 2.1 Instrumentation

Experimental measurements were performed in conjunction with SPICE over the 2013/14 and 2014/15 winter periods
(November 1 to April 30) at the Centre for Atmospheric Research Experiments (CARE) site in Egbert, Ontario, Canada.
Measurements of precipitation accumulation were performed using 600 mm capacity Geonor T-200B3 gauges in unshielded
and reference DFAR configurations. Both gauges were securely mounted on concrete foundations to limit wind-induced
vibrations. The performance of these gauges was confirmed by full-scale field verifications at the start and end of testing, with
annual maintenance to inspect, clean, level, and recharge each gauge. The gauges were charged with a mixture of antifreeze





(60% methanol and 40% propylene glycol) and oil (Esso Bayol 35 in 2013/14, discontinued; Exxon Mobil Isopar M in 2014/15).

Measurements of precipitation occurrence were obtained using a Thies Laser Precipitation Monitor (LPM) installed inside the inner fence of the DFAR. Wind speed and direction measurements at 2 m gauge height were performed with a Vaisala WS425

ultrasonic wind sensor adjacent to the unshielded gauge. Temperature was measured with a Yellow Springs International model 44212 thermistor in an aspirated Stevenson screen. Further details are available in the SPICE final report (Nitu et al., 2018).

## 2.2 Sampling, quality control, and precipitation event selection

The instruments were sampled using a Campbell Scientific CR3000 data logger. For each Geonor T-200B3 precipitation gauge, the frequency and precipitation accumulation for each of the three transducers was reported at 6-second intervals, the latter

computed from the former using manufacturer-provided calibration coefficients. Minutely measurements of precipitation occurrence from the Thies LPM were recorded. The scalar average wind speed and vector average wind direction were recorded over 1-minute intervals. Based on SPICE procedures, these data were processed using a format check to replace missing data with null values, a range check to identify and remove outliers outside the manufacturer-specified output thresholds, a jump filter to remove spikes exceeding maximum point-to-point variation thresholds, and a Gaussian filter to

smooth out high frequency noise in Geonor precipitation accumulation measurements (Nitu et al., 2018). Periods of instrument maintenance and power outages were removed from the analysis. The Geonor accumulation data were aggregated to 1-minute intervals for subsequent analysis.

Precipitation events were identified during both measurement periods using the SPICE event selection procedure (Nitu et al. 2018). These events were defined as 30-minute periods with at least 0.25 mm of precipitation recorded by the reference DFAR

precipitation gauge and at least 60% precipitation occurrence reported by the Thies LPM. The use of the LPM as a secondary confirmation of precipitation occurrence minimizes the likelihood of events with false precipitation due to dumps of snow or ice into the gauge, wind induced vibrations, or other factors. Following the approach of Kochendorfer (2018), a minimum 0.075 mm accumulation threshold was applied for the unshielded gauge to ensure that measurements exceeded the gauge uncertainty and that derived collection efficiency values were reliable. The 30-minute event duration was chosen to be

sufficiently long to reduce noise and ensure high confidence in measured parameters and sufficiently short to avoid the influence of diurnal temperature variations, while also providing a larger number of events for analysis relative to longer durations. Note that unless otherwise stated, all precipitation events referred to hereafter are 30-minute events derived using the above approach.

## 2.3 POSS fall velocity and precipitation type

The POSS is a small upward-facing bistatic X band radar capable of measuring the precipitation fall velocity based on the Doppler frequency shift of the received signal (Canada, 1995;Sheppard, 1990, 2007;Sheppard et al., 1995;Sheppard and Joe, 1994, 2000, 2008). During periods of precipitation, the POSS outputs both the mean and mode received signal frequency



derived from the Doppler frequency spectrum over the previous minute. The mean precipitation fall velocity $u_{f\_mean}$ is estimated from the transmitted wavelength $\lambda$ and the mean frequency $f_{mean}$ of the measured Doppler power density spectrum

for falling precipitation hydrometeors.

$$u_{f\_mean} = \frac{f_{mean}\lambda}{2} , \tag{1a}$$

The mode precipitation fall velocity $u_{f\_mode}$ is described by a similar function, based on the mode frequency $f_{mode}$ of the measured Doppler power density spectrum.

$$u_{f\_mode} = \frac{f_{mode}\lambda}{2} , \tag{1b}$$

For each 30-minute event, the mean and mode event fall velocity correspond to the average of all minutely mean and mode values, respectively. The transfer functions presented in this work were derived using both forms of event fall velocity and assessed in terms of the RMSE and bias error (BE) of adjusted measurements relative to the DFAR. The specific fall velocity indicated for each transfer function corresponds to that which produced the lowest RMSE and BE. The POSS also provides a minutely precipitation type output corresponding to very light, light, moderate, and heavy precipitation for rain, snow, hail,

and undefined precipitation. Each event is classified as 'rain' or 'snow', corresponding to a minimum 70 % occurrence of that precipitation type over the event period (i.e. at least 21 minutes of precipitation occurrence). 'Mixed' precipitation events correspond to the presence of both 'rain' and 'snow' for the remaining events not classified as rain or snow. 'Undefined' precipitation corresponds to events where the precipitation is not captured by the three other classifications.

**2.4 Transfer functions**

Due to the systematic error associated with gauge undercatch, the unshielded gauge can capture less precipitation than the true amount falling in the air. The measured collection efficiency $CE_m$ is defined as the ratio of the precipitation accumulation reported by the unshielded gauge $h_{un}$ relative to that reported by the DFAR $h_{DFAR}$ for each event, and is given by:

$$CE_m = \frac{h_{un}}{h_{DFAR}} , \tag{2}$$

Assuming that the gauge measurement uncertainties are independent and random with equivalent accumulations

(corresponding to a collection efficiency equal to 1) and uncertainties, the uncertainty in the collection efficiency $\sigma_{CE}$ scales with the relative magnitude of the gauge uncertainty $\sigma_h$ and the event accumulation value $h$ by error propagation.

$$\sigma_{CE} = \frac{\sqrt{2}\sigma_h}{h} , \tag{3}$$

Collection efficiency transfer functions $CE$ attempt to capture the performance of the unshielded gauge relative to the reference configuration based on wind speed, temperature, or other meteorological parameters. They can then be applied to





adjust precipitation accumulations from an unshielded gauge in operational settings where reference measurements are not available.

$$h_{\mathrm{adj}} = \frac{h_{\mathrm{un}}}{CE}, \tag{4}$$

Building upon previous work on transfer functions (Goodison, 1978;Goodison et al., 1998;Wolff et al., 2015), Kochendorfer et al. (2017a;2018) used SPICE measurement data from eight test sites to develop an exponential and trigonometric transfer function based on wind speed $u_{\mathrm{w}}$ and air temperature $T$. This is referred to as K$_{\mathrm{Universal}}$ in this work (Eq. 5a). For wind speeds above a threshold value $u_{\mathrm{wt}}$ of 7.2 m s$^{-1}$, the wind speed is fixed at the threshold value (Eq. 5b) to avoid the potential for erroneous catch efficiency values at higher wind speeds that were not well represented in the SPICE measurement dataset. Based on a similar rationale, no adjustment is applied for temperatures above 5 °C. Note that while Kochendorfer et al. (2017b) considered wind speeds at both gauge height and at 10 m, $u_{\mathrm{w}}$ will denote the gauge height wind speed in this work.

$$CE_{\mathrm{K}} \left( u_{\mathrm{w}} \leq u_{\mathrm{wt}}, T \right) = \exp\left[ -b_1 u_{\mathrm{w}} \left( 1 - \tan^{-1}\left( b_2 T \right) + b_3 \right) \right], \tag{5a}$$

$$CE_{\mathrm{K}} \left( u_{w} > u_{\mathrm{wt}}, T \right) = \exp\left[ -b_1 u_{\mathrm{wt}} \left( 1 - \tan^{-1}\left( b_2 T \right) + b_3 \right) \right], \tag{5b}$$

The coefficients for K$_{\mathrm{Universal}}$ are provided in Table 1.

**Table 1.** Unshielded Geonor T-200B3 precipitation gauge collection efficiency transfer function coefficients for solid and mixed precipitation with 30-minute scalar mean wind speed $u_{\mathrm{w}}$ at gauge height for: K$_{\mathrm{Universal}}$ function with wind speed and air temperature $T$ dependence, with constant value above wind speed threshold with Kochendorfer et al. (2017a) coefficients; K$_{\mathrm{CARE}}$ function with wind speed and air temperature dependence, with constant value above wind speed threshold; present study CFD model with dependence on wind speed and mode hydrometeor fall velocity u$_{\mathrm{f\_mode}}$; HE1 model with dependence on wind speed and mean hydrometeor fall velocity u$_{\mathrm{f\_mean}}$ threshold; and HE2 model with wind speed and mode hydrometeor fall velocity dependence and mode hydrometeor fall velocity threshold.

| Description | Eq. | Function | Coefficients | | | | Threshold |
| | | | $b_1$ | $b_2$ | $b_3$ | $b_4$ | |
| --- | --- | --- | --- | --- | --- | --- | --- |
| K$_{\mathrm{Universal}}$ | 5 | $f(u_{\mathrm{w}},T)$ | 0.0785 | 0.729 | 0.407 | - | $u_{\mathrm{wt}} = 7.2$ m s$^{-1}$, $T \leq 5$ °C |
| K$_{\mathrm{CARE}}$ | 5 | $f(u_{\mathrm{w}},T)$ | 0.1651 | 0.186 | -0.757 | | $u_{\mathrm{wt}} = 7.2$ m s$^{-1}$, $T \leq 1.33$ °C |
| CFD | 6 | $f(u_{\mathrm{w}},u_{\mathrm{f\_mode}})$ | 0.908 | 1.387 | 0.143 | 2.422 | $u_{\mathrm{w}} \leq u_{\mathrm{wc}}$, $u_{\mathrm{w}} \leq 10$ m s$^{-1}$ |
| HE1 | 7 | $f(u_{\mathrm{w}},u_{\mathrm{f\_mean}})$ | 0.139 | - | - | - | $u_{\mathrm{f\_mean}} \leq 1.93$ m s$^{-1}$, $u_{\mathrm{w}} \leq 7.19$ m s$^{-1}$ |
| HE2 | 8 | $f(u_{\mathrm{w}},u_{\mathrm{f\_mode}})$ | 0.244 | 0.0869 | - | - | $u_{\mathrm{f\_mode}} \leq 2.81$ m s$^{-1}$ |

Using the same formulation, a site-specific transfer function based on wind speed and temperature was derived using the CARE dataset, for comparison with K$_{\mathrm{Universal}}$. Best-fit regression coefficients were determined by varying the temperature threshold below 5 °C with the collection efficiency constrained to 1 above the threshold value. Solving Eq. 5a for the temperature when the collection efficiency equals 1 provides additional constraint on the $b_3$ coefficient as a function of the $b_2$ coefficient and temperature threshold $T_t$.

$$b_3 = \tan^{-1}\left( b_2 T_t \right) - 1, \tag{5c}$$



The coefficients for the CARE site-specific transfer function, referred to as $K_{CARE}$ in this work, are provided in Table 1. The transfer function derived from CFD simulations in Part I was formulated using the wind speed and hydrometeor fall velocity $u_f$. The collection efficiency decreases nonlinearly with increasing wind speed, but increases with increasing $u_f$ (Eq. 6a). A wind speed cut-off value $u_{wc}$, which is a function of $u_f$, defines the wind speed above which the CE is zero (Eq. 6b). For wind speeds above the cut-off value, the collection efficiency is equal to 0 (Eq. 6c). These expressions were derived using CFD analysis for wind speeds up to 10 m s$^{-1}$. The hydrometeor fall velocity is given by the mode of the POSS Doppler velocity spectrum.

$$CE_{CFD}\left(u_w \leq u_{wc}, u_f\right) = 1 - b_1 \exp\left(-b_2 u_f\right) u_w + b_3 \exp\left(-b_4 u_f\right) u_w^2 , \tag{6a}$$

$$u_{wc} = \frac{b_1}{2b_3} \exp\left[\left(b_4 - b_2\right)u_f\right] - \frac{\sqrt{b_1^2 \exp\left(-2b_2 u_f\right) - 4b_3 \exp\left(-b_4 u_f\right)}}{2b_3 \exp\left(-b_4 u_f\right)} , \tag{6b}$$

$$CE_{CFD}\left(u_w > u_{wc}, u_f\right) = 0 , \tag{6c}$$

## 3 Results

### 3.1 Precipitation type

Using the minutely POSS precipitation type output, events were classified as 'rain', 'snow', 'mixed', or 'undefined' following the methodology in Sect. 2.3. The relative occurrence of different precipitation types as reported by the POSS for the event dataset is summarized in Table 2.

**Table 2.** Mean fall velocities and temperatures of precipitation events by type classification.

| Precipitation phase | Fall velocities (m s$^{-1}$) | Temperatures (°C) | Events (#) |
|---|---|---|---|
| Snow | 0.93 to 2.32 | < 0.5 | 233 |
| Mixed | 1.2 to 4.6 | -7.0 to 2.1 | 45 |
| Undefined | 1.0 to 4.3 | -5.4 to 6.6 | 40 |
| Rain | 1.4 to 6.4 | -4.8 to 18.9 | 196 |

Based on the mean fall velocities and temperatures for each precipitation event (Fig. 1, Table 2), snow events occurred at temperatures below 0.5 °C and with fall velocities of 0.93 m s$^{-1}$ to 2.32 m s$^{-1}$. Mixed events were characterized by mean temperatures between -7.0 °C and 2.1 °C and mean fall velocities between 1.2 m s$^{-1}$ and 4.6 m s$^{-1}$, while undefined precipitation events occurred at mean temperatures between -5.4 °C and 6.6 °C and fall velocities between 1.0 m s$^{-1}$ and 4.3 m s$^{-1}$. Rain events were characterized by mean temperatures between -4.8 °C and 18.9 °C and mean fall velocities between 1.4 m s$^{-1}$ and 6.4 m s$^{-1}$. Over the temperature range between -5 °C and 2 °C, rain, snow, mixed, and undefined precipitation types were all





present, demonstrating the challenge of estimating precipitation type using temperature alone (e.g. as done for the $K_{Universal}$ and $K_{CARE}$ transfer functions). Within this temperature range, a wide variety of mean fall velocities, between 1 and 6 m s$^{-1}$, is also apparent.

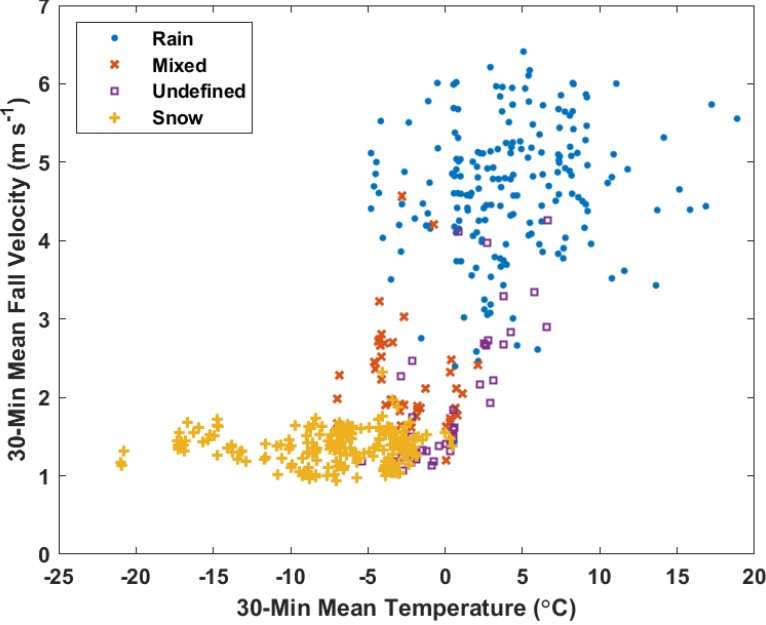

**Figure 1.** Mean air temperature and fall velocity for 30-minute events with rain, snow, mixed, and undefined precipitation (see Table 1 for summary).

**3.2 Collection efficiency**

The unshielded gauge collection efficiency results are shown as a function of the 30-minute DFAR event accumulations in Fig. 2a and stratified by precipitation type classification. The collection efficiency for rain shows less scatter and less uncertainty for higher reference precipitation accumulations. The dashed lines in Fig. 2a show the decrease in the collection efficiency uncertainty with increasing precipitation accumulation for a collection efficiency equal to 1 and a precipitation accumulation uncertainty of 0.1 mm (k = 2) given by Eq. 3. These lines appear to capture the overall trend observed for rain events. The snowfall events show a markedly different trend, however, with collection efficiencies as low as 0.3.

The collection efficiency for all events as a function of mean wind speed and precipitation type classification is shown in Fig. 2b. For rain events, the collection efficiencies are close to 1. For snow, an approximately linear decrease in the collection efficiency with mean wind speed is apparent, with the collection efficiency decreasing to 0.3 at a wind speed of 5 m s$^{-1}$. Mixed precipitation collection efficiencies span a range of values between those of rain and snow. For undefined precipitation, some





events have collection efficiencies close to 1 at high wind speeds, similar to rain events, while others appear to decrease with increasing wind speed in a similar fashion to that observed for snow events.

**Figure 2.** Collection efficiency of the unshielded gauge as a function of: (a) precipitation accumulation and event precipitation type (dashed
lines illustrate accumulation uncertainty threshold); (b) wind speed and event precipitation type; (c) wind speed and mean air temperature $T$ categories; and (d) wind speed and mode fall velocity $u_p$ categories.



The dependence of collection efficiencies on the mean wind speed over four separate mean temperature ranges is shown in Fig. 2c. For mean event temperatures above 2 °C, the collection efficiencies are generally close to 1, typical of rain. For

temperatures between -5 °C and -2 °C and between -2 °C and 2 °C, a range of collection efficiency values are observed, from those typical of snow to those typical of rain. This variation is attributed to the wide range of fall velocities within this temperature range, which includes snow, rain, and mixed precipitation events (Fig. 2). At colder temperatures, below -5 °C, collection efficiencies appear to decrease approximately linearly with wind speed, consistent with the trend observed for snow events in Fig. 2b.

Stratifying the collection efficiency results as a function of mean event wind speed by the mode fall velocity shows more distinct trends (Fig. 2d) relative to those observed when stratifying by temperature (Fig. 2c). Collection efficiencies are close to 1 for fall velocities greater than 2.5 m s$^{-1}$, generally corresponding to rain. Conversely, fall velocities below 1.5 m s$^{-1}$ show an approximately linear decrease in collection efficiency with increasing wind speed up to about 6 m s$^{-1}$. A number of the values with higher collection efficiencies in this low fall velocity range correspond to mixed precipitation, where both snow and rain may be present. Between 1.5 m s$^{-1}$ to 2.5 m s$^{-1}$ fall velocity, intermediate collection efficiency values are evident, with collection efficiencies transitioning from lower to higher values, despite a fewer number of observations in this range.

The dependence of the collection efficiency on 30-minute mean air temperature and the 30-minute mean of mode fall velocity values for 2 m s$^{-1}$ to 4 m s$^{-1}$ wind speeds is shown in Figs. 3a and 3b, respectively. This range of wind speeds shows the most significant overlap among rain, snow, mixed and undefined precipitation, and is sufficiently high that a wide range of collection

efficiencies is observed. The collection efficiency generally increases with temperature (Fig. 3a), although values are spread broadly across air temperature values, with collection efficiencies below 0.6 occurring between -22 °C and 0 °C. More clearly-defined trends with fall velocity are apparent, with collection efficiencies increasing sharply with fall velocity up to ~ 2.5 m s$^{-1}$ and distributed around 1 for higher fall velocities.





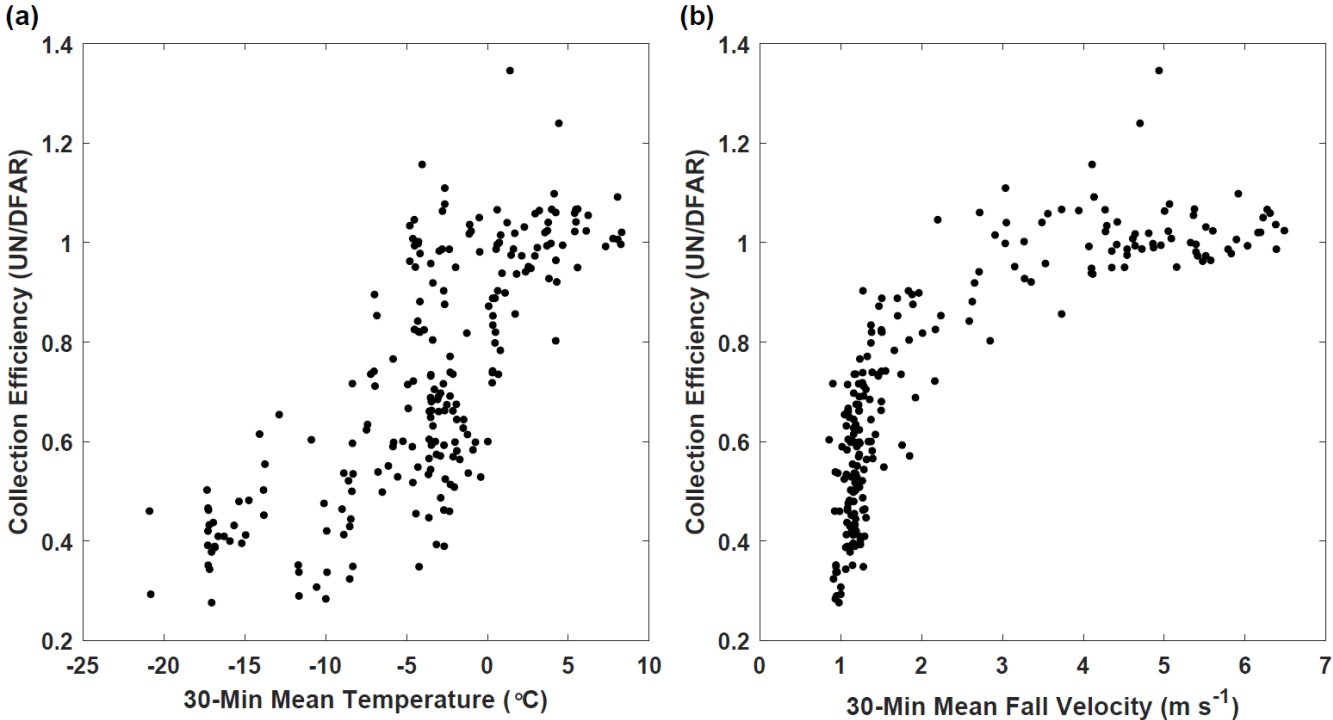

Figure 3. Collection efficiency of unshielded gauge for 2 m s$^{-1}$ to 4 m s$^{-1}$ 30-minute mean wind speeds with (a) 30-minute mean air temperature and (b) 30-minute mean of mode fall velocities.

### 3.3 Derivation of fall velocity transfer functions from CE results

Two additional transfer functions were formulated based on the apparent linear dependence of CE on wind speed for different hydrometeor fall velocity regimes observed in experimental results (Fig. 2d). These functions are applicable to all hydrometeor types, and have different fall velocity thresholds to describe the transition of precipitation phase from the lower fall velocities characteristic of snow to the higher fall velocities characteristic of rain and mixed precipitation.

The first transfer function, referred to as HE1, is based on the assumption of a linear decrease in collection efficiency $CE_{\mathrm{HE1}}$ with wind speed $u_{\mathrm{w}}$ for hydrometeors with mean fall velocity $u_{\mathrm{f}}$ below 1.93 m s$^{-1}$, generally corresponding to snowfall. This linear decrease is extrapolated to 7.19 m s$^{-1}$ wind speed (Eq. 7a), above which the collection efficiency for snowfall is zero (Eq. 7b). For hydrometeors with mean fall velocity greater than 1.93 m s$^{-1}$, corresponding to mixed and liquid precipitation, the collection efficiency is 1 (Eq. 7c).

$$CE_{\mathrm{HE1}}\left(u_{\mathrm{w}} \leq 7.19\mathrm{m\,s}^{-1}, u_{\mathrm{f}} \leq 1.93\mathrm{m\,s}^{-1}\right) = 1 - b_{1}u_{\mathrm{w}}, \tag{7a}$$

$$CE_{\mathrm{HE1}}\left(u_{\mathrm{w}} > 7.19\mathrm{m\,s}^{-1}, u_{\mathrm{f}} \leq 1.93\mathrm{m\,s}^{-1}\right) = 0, \tag{7b}$$





$$CE_{\mathrm{HE1}}\left(u_{\mathrm{f}} > 1.93\mathrm{m\ s}^{-1}\right) = 1,\qquad(7c)$$

The second transfer function, referred to as HE2, adds another dimension to describe the slope of the linear decrease in CE with increasing wind speed: the hydrometeor fall velocity. For mode fall velocity $u_{\mathrm{f}}$ below 2.81 m s$^{-1}$ and wind speed $u_{\mathrm{w}}$ below the cut-off value, which is also dependent on the fall velocity, the collection efficiency $CE_{\mathrm{HE2}}$ is assumed to decrease linearly with decreasing wind speed to zero (Eq. 8a). For mode fall velocity below 2.81 m s$^{-1}$ and wind speed above the cut-off value, the collection efficiency is zero (Eq. 8b). For mode fall velocity above 2.81 m s$^{-1}$, the collection efficiency is equal to 1 (Eq. 8c).

$$CE_{\mathrm{HE2}}\left(u_{\mathrm{w}} \le \frac{1}{b_1 - b_2 u_{\mathrm{f}}}, u_{\mathrm{f}} \le 2.81\mathrm{m\ s}^{-1}\right) = 1 - \left(b_1 - b_2 u_{\mathrm{f}}\right)u_{\mathrm{w}},\qquad(8a)$$

$$CE_{\mathrm{HE2}}\left(u_{\mathrm{w}} > \frac{1}{b_1 - b_2 u_{\mathrm{f}}}, u_{\mathrm{f}} \le 2.81\mathrm{m\ s}^{-1}\right) = 0,\qquad(8b)$$

$$CE_{\mathrm{HE2}}\left(u_{\mathrm{f}} > 2.81\mathrm{m\ s}^{-1}\right) = 1,\qquad(8c)$$

## 3.4 Assessment of transfer functions: collection efficiency

Observed collection efficiencies were compared with adjusted values using both existing transfer functions from SPICE and those presented in this work. Results are presented in Fig. 4, with relevant transfer function parameters compiled in Table 1, and resulting bias errors, root mean square errors, and correlation coefficients (r) presented in Table 3. To further contextualize the assessment of the different transfer functions, the RMSE results are presented for different precipitation classifications, temperature ranges, and fall velocity ranges in Tables 4 to 6, respectively.

**Figure 4.** Collection efficiency of unshielded gauge as a function of wind speed for: (a) mean air temperature $T$ categories for the $K_{Universal}$ and $K_{CARE}$ transfer functions; (b) mode fall velocity $u_p$ categories with the CFD transfer function; (c) mean fall velocity $u_p$ categories for the HE1 transfer function; and (d) mode fall velocity $u_p$ categories with the HE2 transfer function.



**Table 3.** Unshielded gauge 30-minute event bias error (BE), root mean square error (RMSE), correlation coefficient (r), and number of events (N) for collection efficiency and precipitation accumulation between the unshielded and reference DFAR shielded Geonor T-200B3 gauge for: unadjusted comparison; $K_{Universal}$ transfer function with wind speed and air temperature dependence; $K_{CARE}$ transfer function with wind speed and air temperature dependence; present study CFD transfer function with wind speed and mode fall velocity dependence; HE1 transfer function with wind speed and mean fall velocity dependence; and HE2 transfer function with wind speed and mode fall velocity dependence. Statistics are based on the comparison of experimental results from the CARE site between November 1 and April 30, 2013/14 and 2014/15.

| | Collection efficiency | | | Precip accum (mm) | | | |
|---|---|---|---|---|---|---|---|
| Description | BE | RMSE | r | BE | RMSE | r | N |
| Unadjusted | - | - | - | -0.13 | 0.24 | 0.900 | 514 |
| $K_{Universal}$ | 0.07 | 0.15 | 0.853 | 0.07 | 0.20 | 0.949 | 514 |
| $K_{CARE}$ | -0.005 | 0.12 | 0.878 | 0.002 | 0.13 | 0.963 | 514 |
| CFD | -0.02 | 0.08 | 0.949 | 0.011 | 0.08 | 0.986 | 514 |
| HE1 | 0.0004 | 0.10 | 0.928 | 0.006 | 0.09 | 0.983 | 514 |
| HE2 | -0.009 | 0.08 | 0.950 | 0.006 | 0.07 | 0.988 | 514 |

**Table 4.** Unshielded gauge 30-minute event collection efficiency RMSE between the unshielded and reference DFAR shielded Geonor T-200B3 gauge by POSS precipitation type for: $K_{Universal}$ transfer function with wind speed and air temperature dependence; $K_{CARE}$ transfer function with wind speed and air temperature dependence; present study CFD transfer function with wind speed and mode fall velocity dependence; HE1 transfer function with wind speed and mean fall velocity dependence; and HE2 transfer function with wind speed and mode fall velocity dependence. Statistics are based on the comparison of experimental results from the CARE site between November 1 and April 30, 2013/14 and 2014/15.

| | RMSE | | | |
|---|---|---|---|---|
| | Rain | Mixed | Undefined | Snow |
| Description | (N = 196) | (N = 45) | (N = 40) | (N = 233) |
| $K_{Universal}$ | 0.17 | 0.27 | 0.09 | 0.09 |
| $K_{CARE}$ | 0.12 | 0.20 | 0.13 | 0.11 |
| CFD | 0.08 | 0.09 | 0.09 | 0.09 |
| HE1 | 0.07 | 0.16 | 0.08 | 0.10 |
| HE2 | 0.08 | 0.10 | 0.09 | 0.08 |

**Table 5.** Unshielded gauge 30-minute event collection efficiency RMSE between the unshielded and reference DFAR shielded Geonor T-200B3 gauge by temperature classification for: $K_{Universal}$ transfer function with wind speed and air temperature dependence; $K_{CARE}$ transfer function with wind speed and air temperature dependence; present study CFD transfer function with wind speed and mode fall velocity dependence; HE1 transfer function with wind speed and mean fall velocity dependence; and HE2 transfer function with wind speed and mode fall velocity dependence. Statistics are based on the comparison of experimental results from the CARE site between November 1 and April 30, 2013/14 and 2014/15.

| | RMSE | | | |
|---|---|---|---|---|
| | $T > 2\ °C$ | $-2\ °C < T \leq 2\ °C$ | $-5\ °C < T \leq -2\ °C$ | $T \leq -5\ °C$ |
| Description | (N = 150) | (N = 89) | (N = 134) | (N = 141) |
| $K_{Universal}$ | 0.08 | 0.19 | 0.21 | 0.11 |
| $K_{CARE}$ | 0.07 | 0.13 | 0.17 | 0.10 |
| CFD | 0.09 | 0.08 | 0.08 | 0.09 |
| HE1 | 0.07 | 0.10 | 0.11 | 0.10 |
| HE2 | 0.09 | 0.08 | 0.07 | 0.08 |





**Table 6.** Unshielded gauge 30-minute event collection efficiency RMSE between the unshielded and reference DFAR shielded Geonor T-200B3 gauge by fall velocity classification for: $K_{Universal}$ transfer function with wind speed and air temperature dependence; $K_{CARE}$ transfer function with wind speed and air temperature dependence; present study CFD transfer function with wind speed and mode fall velocity dependence; HE1 transfer function with wind speed and mean fall velocity dependence; and HE2 transfer function with wind speed and mode fall velocity dependence. Statistics are based on the comparison of experimental results from the CARE site between November 1 and April 30, 2013/14 and 2014/15.

| | RMSE | | | |
|---|---|---|---|---|
| | $u_f > 2.5$ m s$^{-1}$ | 2 m s$^{-1} < u_f \leq 2.5$ m s$^{-1}$ | 1.5 m s$^{-1} < u_f \leq 2$ m s$^{-1}$ | $u_f \leq 1.5$ m s$^{-1}$ |
| Description | (N = 212) | (N = 15) | (N = 40) | (N = 247) |
| $K_{Universal}$ | 0.19 | 0.23 | 0.16 | 0.09 |
| $K_{CARE}$ | 0.13 | 0.17 | 0.12 | 0.11 |
| CFD | 0.08 | 0.10 | 0.08 | 0.09 |
| HE1 | 0.08 | 0.13 | 0.15 | 0.10 |
| HE2 | 0.08 | 0.12 | 0.08 | 0.08 |

Both $K_{Universal}$ and the site-specific $K_{CARE}$ transfer function have continuous temperature dependence and display similar profiles at -8 °C, with the collection efficiency for the $K_{CARE}$ transfer function decreasing more gradually with wind speed compared to the $K_{Universal}$ transfer function at -4 °C and 0 °C (Fig. 4a). Using the approach outlined in Sect. 2.4, a temperature cutoff $T_t$ of 1.33 °C for the best-fit $K_{CARE}$ transfer function was found to minimize the precipitation accumulation RMSE. The overall collection efficiency root mean square error is reduced from 0.15 for the $K_{Universal}$ transfer function to 0.12 for the $K_{CARE}$ transfer function (Table 3). The bias error is also reduced from 0.07 for the $K_{Universal}$ transfer function to -0.005 for the best-fit $K_{CARE}$ transfer function. For $K_{Universal}$ and $K_{CARE}$, respectively, the RMSE is reduced from 0.17 to 0.12 for rain and from 0.27 to 0.20 for mixed precipitation, with slightly elevated RMSE from 0.09 to 0.13 for undefined precipitation and 0.09 to 0.11 for snow (Table 4). For mean event temperatures between -2 °C and 2 °C, and between -5 °C and -2 °C, respectively, the RMSE values of 0.19 and 0.21 for the $K_{Universal}$ transfer function are relatively large compared to the 0.13 and 0.17 values for the $K_{CARE}$ transfer function (Table 5). This results from the more gradual decrease in the $K_{CARE}$ transfer function with wind speed over these temperature ranges (Fig. 4a).

A comparison of the CFD transfer function with observed CE is shown in Fig. 4b. Overall, the measured data have less scatter when stratified by fall velocity than when stratified by temperature (Table 3, Figs. 4a and b). The CFD transfer function provides a lower overall RMSE (0.08) and higher r (0.949) relative to the $K_{Universal}$ and $K_{CARE}$ transfer functions based on temperature. Reductions in the collection efficiency RMSE using the CFD transfer function are most pronounced for rain and mixed precipitation (Table 4) and for mean event temperatures between -2 °C and 2 °C and between -5 °C and -2 °C (Table 5) compared with the $K_{Universal}$ and $K_{CARE}$ functions. Collection efficiency RMSE values are between 0.08 and 0.10 over all fall velocity classes, despite fewer numbers of events with fall velocities between 1.5 m s$^{-1}$ and 2.5 m s$^{-1}$ (Table 6).

The HE1 transfer function provides good agreement with observed data in the mean fall velocity regimes relevant to snow and rain (Fig. 4c), resulting in an overall RMSE of 0.10, BE of 0.0004, and r of 0.928 (Table 3). The RMSE for mixed precipitation





is 0.16, which is lower than that of the $K_{CARE}$ transfer function with temperature (0.20) but higher that that of the CFD model (0.09), which varies continuously with fall velocity (Table 4).

The HE2 function better captures the observed collection efficiencies for mode fall velocities between the snow and rain regimes (Fig. 4d), improving the overall RMSE to 0.08 and r to 0.95, while increasing slightly the BE (-0.009) relative to HE1 (Table 3). Note the distinction between mean fall velocity for HE1 and mode fall velocity for HE2 (and CFD). In general, the

Doppler frequency spectrum tends to be skewed such that mode fall velocities are slightly lower than the mean fall velocities, impacting the fits to observed data. The HE2 transfer function provides similar results to that of the CFD transfer function, with slightly higher RMSE values for mixed precipitation and slightly reduced RMSE values for snow (Table 4) and temperatures below -2 °C (Table 5). For intermediate fall velocities between 2.0 m s$^{-1}$ and 2.5 m s$^{-1}$, the HE2 transfer function, with a linear change in collection efficiency with fall velocity, has a higher RMSE (0.12) than that for the CFD function (0.10),

which exhibits a nonlinear change in collection efficiency with fall velocity (Table 6). Only 15 events were recorded in this intermediate fall velocity range with higher uncertainty relative to the CFD function. In contrast, 212 events were recorded at fall velocities above 2.5 m s$^{-1}$ and 247 events at fall velocities below 1.5 m s$^{-1}$, representing a greater proportion of the events with lower RMSE relative to the CFD function.

### 3.5 Assessment of transfer functions: precipitation accumulation

The unadjusted and adjusted accumulated precipitation values are compared with reference DFAR accumulation measurements in Fig. 5. Bias, RMSE, and correlation coefficient results are shown in Table 3. Similar to the approach for assessing transfer functions based on collection efficiency results in Sect. 3.4, the precipitation accumulation RMSE results for each transfer function are assessed by precipitation classification, temperature range, and fall velocity range in Tables 7 to 9, respectively.





**Figure 5.** Unshielded and reference DFAR 30-minute event precipitation accumulation comparison for: (a) unadjusted precipitation accumulation; (b) $K_{Universal}$ continuous transfer function with wind speed and air temperature dependence; (c) $K_{CARE}$ continuous transfer function with wind speed and air temperature dependence; (d) CFD transfer function with wind speed and fall velocity dependence; (e) HE1 transfer function with wind speed and fall velocity dependence; and (f) HE2 transfer function with wind speed and fall velocity dependence.





**Table 7.** Unshielded gauge 30-minute event RMSE (mm) between the unshielded and reference DFAR shielded Geonor T-200B3 gauge by POSS precipitation type for: unadjusted comparison; $K_{Universal}$ transfer function with wind speed and air temperature dependence; $K_{CARE}$ transfer function with wind speed and air temperature dependence; present study CFD transfer function with wind speed and mode fall velocity dependence; HE1 transfer function with wind speed and mean fall velocity dependence; and HE2 transfer function with wind speed and mode fall velocity dependence. Statistics are based on the comparison of experimental results from the CARE site between November 1 and April 30, 2013/14 and 2014/15.

| | RMSE (mm) | | | |
|---|---|---|---|---|
| Description | Rain (N = 196) | Mixed (N = 45) | Undefined (N = 40) | Snow (N = 233) |
| Unadjusted | 0.04 | 0.15 | 0.09 | 0.35 |
| $K_{Universal}$ | 0.25 | 0.33 | 0.05 | 0.10 |
| $K_{CARE}$ | 0.14 | 0.22 | 0.06 | 0.11 |
| CFD | 0.04 | 0.07 | 0.04 | 0.11 |
| HE1 | 0.04 | 0.17 | 0.04 | 0.10 |
| HE2 | 0.04 | 0.09 | 0.04 | 0.09 |

**Table 8.** Unshielded gauge 30-minute event RMSE (mm) between the unshielded and reference DFAR shielded Geonor T-200B3 gauge by temperature classification for: unadjusted comparison; $K_{Universal}$ transfer function with wind speed and air temperature dependence; $K_{CARE}$ transfer function with wind speed and air temperature dependence; present study CFD transfer function with wind speed and mode fall velocity dependence; HE1 transfer function with wind speed and mean fall velocity dependence; and HE2 transfer function with wind speed and mode fall velocity dependence. Statistics are based on the comparison of experimental results from the CARE site between November 1 and April 30, 2013/14 and 2014/15.

| | RMSE (mm) | | | |
|---|---|---|---|---|
| Description | $T > 2\ °C$ (N = 150) | $-2\ °C < T ≤ 2\ °C$ (N = 89) | $-5\ °C < T ≤ -2\ °C$ (N = 134) | $T ≤ -5\ °C$ (N = 141) |
| Unadjusted | 0.04 | 0.14 | 0.23 | 0.39 |
| $K_{Universal}$ | 0.05 | 0.25 | 0.29 | 0.12 |
| $K_{CARE}$ | 0.04 | 0.11 | 0.20 | 0.12 |
| CFD | 0.05 | 0.06 | 0.08 | 0.11 |
| HE1 | 0.04 | 0.12 | 0.09 | 0.10 |
| HE2 | 0.05 | 0.07 | 0.08 | 0.09 |

**Table 9.** Unshielded gauge 30-minute event RMSE (mm) between the unshielded and reference DFAR shielded Geonor T-200B3 gauge by fall velocity classification for: unadjusted comparison; $K_{Universal}$ transfer function with wind speed and air temperature dependence; $K_{CARE}$ transfer function with wind speed and air temperature dependence; present study CFD transfer function with wind speed and mode fall velocity dependence; HE1 transfer function with wind speed and mean fall velocity dependence; and HE2 transfer function with wind speed and mode fall velocity dependence. Statistics are based on the comparison of experimental results from the CARE site between November 1 and April 30, 2013/14 and 2014/15.

| | RMSE (mm) | | | |
|---|---|---|---|---|
| Description | $u_f > 2.5\ m\ s^{-1}$ (N = 212) | $2\ m\ s^{-1} < u_f ≤ 2.5\ m\ s^{-1}$ (N = 15) | $1.5\ m\ s^{-1} < u_f ≤ 2\ m\ s^{-1}$ (N = 40) | $u_f ≤ 1.5\ m\ s^{-1}$ (N = 247) |
| Unadjusted | 0.04 | 0.06 | 0.16 | 0.34 |
| $K_{Universal}$ | 0.26 | 0.22 | 0.22 | 0.10 |
| $K_{CARE}$ | 0.15 | 0.14 | 0.15 | 0.11 |
| CFD | 0.04 | 0.05 | 0.06 | 0.10 |
| HE1 | 0.04 | 0.06 | 0.16 | 0.10 |
| HE2 | 0.04 | 0.06 | 0.07 | 0.09 |



In the comparison of unadjusted accumulation measurements with reference values (Fig. 5a), some values fall along the 1-to-
420 1 line, while others are considerably lower. The values along the 1-to-1 line generally correspond to rain events with high
precipitation fall velocity, or to events with low mean wind speeds. The RMSE for the unadjusted unshielded gauge
measurements relative to the DFAR is 0.24 mm, with a bias error of -0.13 mm and correlation coefficient of 0.900 (Table 3).
Using the $K_{Universal}$ transfer function, with wind and temperature dependence, shifts the adjusted values up to and above the 1-
to-1 line (Fig. 5b). This yields a positive bias error of 0.07 mm, reduced RMSE of 0.20 mm, and correlation coefficient of
425 0.949 (Table 3) relative to the unadjusted measurements (Fig. 5a). While the $K_{Universal}$ transfer function greatly reduces the
RMSE for snow from 0.35 mm to 0.10 mm compared with unadjusted values, the RMSE is increased from 0.04 mm to 0.25
mm for rain, and from 0.15 mm to 0.33 mm for mixed precipitation (Table 7). Compared with the unadjusted results, RMSE
increases for the $K_{Universal}$ function are also apparent for temperatures between -2 °C and 2 °C and between -5 °C and -2 °C
(Table 8), and for fall velocities greater than 1.5 m s$^{-1}$ (Table 9).
Applying the site-specific $K_{CARE}$ transfer function, based on the best-fit results to the CARE SPICE dataset, results in a reduced
bias error of 0.002 mm, lower RMSE of 0.13 mm, and higher correlation coefficient of 0.963 (Table 3) relative to the $K_{Universal}$
results, with the scatter in adjusted accumulations more evenly balanced across the 1-to-1 line (Fig. 5c). The scatter in adjusted
values using the $K_{CARE}$ transfer function results primarily from mixed precipitation (Table 7) at temperatures between -5 °C
and -2 °C (Table 8). Compared to the $K_{Universal}$ transfer function, the $K_{CARE}$ transfer function has lower RMSE values for rain
(0.14 mm) and mixed precipitation (0.22 mm), with 0.01 mm higher RMSE for undefined precipitation and snow (Table 7).
The more rapid increase in collection efficiency with temperature for $K_{CARE}$ relative to $K_{Universal}$ reduces the overadjustment of
some of the rain and mixed precipitation events at temperatures between -5 °C and -2 °C, at the expense of the underadjustment
of some snow events in this temperature range. It is also worth noting that the adjusted precipitation accumulation RMSE for
the $K_{CARE}$ transfer function is larger than that for unadjusted results for rain and mixed precipitation, similar to the results for
$K_{Universal}$. Both the $K_{Universal}$ and $K_{CARE}$ transfer functions with temperature show signs of heteroscedasticity, with an increased
spread of values with increasing magnitude of event precipitation accumulation.

Applying the CFD transfer function results in a greatly reduced spread of values about the 1-to-1 line (Fig. 5d). The spread
does not appear to increase with increasing precipitation accumulation. The overall RMSE is reduced to 0.08 mm, 2.5 times
lower than that for the $K_{Universal}$ transfer function, with a bias error of 0.011 mm and correlation coefficient of 0.986 (Table 3).
The RMSE is reduced from 0.25 mm for the $K_{Universal}$ transfer function to 0.04 mm using the CFD transfer function for rain,
and from 0.33 mm to 0.07 mm (4.7 times lower) for mixed precipitation, while RMSE results for undefined precipitation and
snow are within 0.01 mm (Table 7). Reductions in the RMSE using the CFD transfer function compared with the $K_{Universal}$
transfer function are most pronounced for mean event temperatures between -5 °C and 2 °C (Table 8). Over this temperature
range, rain, mixed precipitation, and snow may be present, corresponding to a wide range of fall velocities and collection
efficiencies. The CFD transfer function is better able to distinguish among these precipitation types – and their respective



collection efficiencies – based on its dependence on hydrometeor fall velocity. Across the fall velocity classifications in Table 9, the RMSE using the CFD transfer function increases from 0.04 mm for fall velocities greater than 2.5 m s$^{-1}$ to 0.10 mm for fall velocities less than 1.5 m s$^{-1}$. As shown in Table 9, the RMSE for the CFD transfer function matches the value for unadjusted measurements at fall velocities greater than 2.5 m s$^{-1}$, where collection efficiencies are close to 1. At lower fall velocities, where the bias due to gauge undercatch is more prevalent, the RMSE values for the CFD function are lower than those for the unadjusted measurements.

Using the HE1 transfer function results in similar overall improvement in the agreement between adjusted and DFAR accumulation values as observed for the CFD function (Fig. 5e). The adjusted values appear to be distributed symmetrically about the 1-to-1 line. Furthermore, there is close agreement over the full range of accumulation values; that is, the spread in values does not increase with the magnitude of precipitation accumulation. This results in a lower RMSE of 0.09 mm and a higher correlation coefficient of 0.983 relative to the $K_{CARE}$ transfer function results. While the RMSE for rain (0.04 mm) using the HE1 transfer function is improved compared with the $K_{CARE}$ transfer function results, the RMSE for mixed precipitation is only marginally better (0.17 mm).

Applying the HE2 transfer function provides further improvement, with adjusted accumulation values more tightly clustered around the 1-to-1 line (Fig. 5f). The overall RMSE is 0.07 mm, which is 3.3 times lower than that for the unadjusted unshielded gauge measurements, and 1.8 times lower than the $K_{CARE}$ transfer function based on mean event temperature and wind speed. The HE2 transfer function exhibits the lowest overall RMSE for snow (0.09 mm), with a RMSE of 0.09 mm for mixed precipitation, which is slightly higher than that for the CFD function (0.07 mm), but much lower than that for the $K_{CARE}$ (0.22 mm) and HE1 (0.17 mm) transfer functions. Further, the correlation coefficient of 0.988 is the highest among the transfer functions assessed.

## 4 Discussion

Transfer functions were derived using accumulated precipitation amounts reported by automatic weighing precipitation gauges over 30 minute periods. A 0.25 mm accumulation threshold was applied to reference measurements from a DFAR, corresponding to an average precipitation rate of 0.5 mm/h over 30 minutes. A lower threshold of 0.075 mm was applied to measurements from the unshielded Geonor gauge to ensure collection efficiency estimates were reliable. This approach is consistent with that used in SPICE (Nitu et al., 2018) and the related derivation of transfer functions (Kochendorfer et al., 2017a). While automatic precipitation gauges can report at a temporal resolution of one minute, or even higher, the extension of the transfer function derivation and evaluation to other temporal periods, or different accumulation thresholds, is beyond the scope of this work.

The Kochendorfer et al. (2017a) universal transfer function with wind speed and air temperature dependence, $K_{Universal}$, was derived from measurements at eight SPICE sites in the interest of making the transfer function broadly applicable across different climates. This broad applicability is furthered by the widespread availability of air temperature and wind speed



measurements at meteorological stations. Recent studies have demonstrated that the performance of $K_{Universal}$ can vary substantially by site (Smith et al., 2020). Therefore, site-specific $K_{CARE}$ transfer function coefficients were also derived for comparison in the present study.

The $K_{CARE}$ transfer function has a lower temperature threshold and exhibits larger increases in collection efficiency with increasing temperature relative to $K_{Universal}$ (Fig. 4a). These differences improved the overall RMSE for $K_{CARE}$ by reducing the over-adjustment of some rain and mixed precipitation events; however, this improvement came at the expense of under-adjusting some snow events at warmer temperatures. The use of this approach warrants further study over longer periods to better understand the performance impacts of seasonal variability and assessment at other sites and climate regions with different precipitation characteristics and proportions.

Both the $K_{Universal}$ and $K_{CARE}$ transfer functions performed well for snow, but were limited by their ability to distinguish among snow, rain, and mixed precipitation at temperatures between -5 °C and 2 °C. The largest uncertainties in collection efficiency and adjusted accumulation estimates were observed over this temperature range. Adjustments using wind speed and hydrometeor fall velocity, however, addressed this shortcoming and provided improved collection efficiency and adjusted accumulation estimates. The CFD transfer function, derived from time-averaged numerical simulation results over a wide range of wind speeds and hydrometeor fall velocities, resulted in low RMSE values overall and across rain, snow, mixed, and undefined precipitation types. These results demonstrate the fundamental importance of both wind speed and hydrometeor fall velocity on gauge collection efficiency predicted by the model results of Part I and earlier studies (Nešpor and Sevruk, 1999;Thériault et al., 2012). This transfer function exhibited the lowest RMSE of all transfer functions for mixed precipitation and for intermediate fall velocities between 1.5 m s$^{-1}$ to 2.5 m s$^{-1}$, which is attributed to its nonlinear increase in collection efficiency with fall velocity. As this transfer function was derived theoretically, it is applicable across different sites and climate regimes with different types and relative proportions of hydrometeors. The present results also support the methodology for the CFD model, which can be extended to other shield and gauge combinations.

The HE1 transfer function showed good results for snow, supporting its use for the unshielded gauge. This approach is straightforward to implement based on its simplicity, and is less reliant on the accuracy of fall velocity estimates beyond the fall velocity cutoff. The collection efficiency for the HE1 transfer function decreases to zero at a wind speed of 7.19 m s$^{-1}$. This demonstrates the limitation of adjusting unshielded gauge snow measurements at windy sites, where the 30-minute mean wind speeds exceed the cutoff value and the captured accumulations are small relative to gauge uncertainties. The latter can lead to large uncertainty in adjusted measurements, as demonstrated by other studies applying transfer functions to unshielded gauge measurements at windy sites (Smith et al., 2020). The numerical results in Part I suggest a more gradual decrease in collection efficiency at higher wind speeds compared with the HE1 transfer function, as some hydrometeors with higher fall velocities are still able to be captured by the gauge; however, these accumulations remain small relative to gauge uncertainties, particularly in windy conditions, making them difficult to assess. The use of shielding or gauges with higher sensitivity could extend the applicability of this approach for use at windy sites.





A limitation of the HE1 transfer function is the minimal improvement in the RMSE for mixed precipitation and fall velocities between 1.5 m s$^{-1}$ to 2.0 m s$^{-1}$ relative to the $K_{CARE}$ function. This is due to the over-adjustment of mixed precipitation events with fall velocities slightly below the cutoff value, and the under-adjustment of mixed precipitation events with fall velocities slightly above the cutoff. While the RMSE for mixed precipitation is still lower than that for adjustments based on temperature

and wind speed ($K_{Universal}$, $K_{CARE}$), further improvements are obtained by using transfer functions with continuous fall velocity dependence; specifically, the CFD and HE2 transfer functions.

The HE2 transfer function, with a linear increase in collection efficiency with fall velocity, yields a greater reduction in the RMSE for mixed precipitation relative to the HE1 transfer function. The HE2 transfer function results show a higher RMSE for mixed precipitation than those for the CFD function, possibly due to the nonlinearity in the latter with fall velocity. The

HE2 transfer function, however, yields the best RMSE results for snow, temperatures below -5 °C, and fall velocities below 1.5 m s$^{-1}$. Adjusted uncertainties for snow are approximately two times higher than those for rain, and show similar trends with increasing temperature and decreasing fall velocity. The former may be due to the lower event accumulations for snow relative to rain, with measured values in closer proximity to the gauge uncertainty. While this transfer function was derived using the CARE dataset, it is more universally applicable than adjustments based on temperature, for which the relative proportions of

rain, snow, and mixed precipitation at warmer temperatures can influence fit results. Further testing at other sites is recommended to assess this in different climate regions, with different hydrometeor types and associated fall velocities.

It is evident that the performance of catchment-type precipitation gauges is dependent on wind speed and the aerodynamic properties of both the gauge and incident hydrometeors (Nešpor and Sevruk, 1999;Thériault et al., 2012;Colli et al., 2016b). Part I of this study demonstrated this dependence from a theoretical perspective, resulting in a transfer function that

incorporates hydrometeor fall velocity. The present contribution validated this approach, which resulted in improved precipitation estimates from an unshielded gauge relative to those using surface temperature as a proxy for precipitation phase or type. Indeed, the use of surface temperature in this manner can be instructive, but does not capture the conditions defining hydrometeor initiation and growth aloft (Kienzle, 2008;Harder and Pomeroy, 2013;Thériault et al., 2012).

In this study, the fall velocity of hydrometeors reported by the POSS provided direct measurement of a key parameter related

to the aerodynamics of the catchment process. In Canada, the POSS was deployed operationally to report present weather as part of an automatic weather station. Globally, other types of disdrometers (e.g. OTT Parsivel[2], Thies Laser Precipitation Monitor) have been deployed operationally and can also provide hydrometeor vertical velocities. The uncertainty in fall velocity estimates for different technologies, hydrometeor types, sizes, fall velocities, wind speeds, and wind directions remains to be assessed. These sensors can also be useful for reporting present weather and verifying the occurrence of

precipitation based on their high sensitivity (Nitu et al., 2018;Sheppard and Joe, 2000).

The results from this study demonstrate that the combined use of accumulation reports from an unshielded weighing gauge with fall velocities reported by a disdrometer, wind speed measurements, and an appropriate transfer function can greatly reduce the uncertainty of precipitation accumulation measurements. At high wind speeds (> 7 m s$^{-1}$), the unshielded gauge catch may be insufficient for adjustment due to the low measured quantities. The extension of the approach in the present study





to shielded precipitation gauges or gauge designs with higher sensitivity may provide a means of reducing the measurement uncertainty for automatic gauges in windy environments. Application to light snow events and different event durations are other areas for future study.

## 5 Conclusions

Three collection efficiency transfer functions with gauge height wind speed and precipitation fall velocity dependence are
presented for unshielded Geonor T-200B3 precipitation gauges and compared to universal and site-specific transfer functions with wind speed and temperature dependence. These functions employ different models to adjust precipitation accumulation measurements for wind-induced undercatch, including:

(1) The nonlinear CFD transfer function model presented in Part I, with collection efficiency decreasing with wind speed and increasing with precipitation fall velocity.

(2) The HE1 transfer function, with a linear decrease in collection efficiency with wind speed for 30-minute mean fall velocity below 1.93 m s$^{-1}$, and a collection efficiency of 1 above this value.

(3) The HE2 transfer function, with the linear wind speed dependence transitioning with increasing mode fall velocity to provide a collection efficiency of 1 when the mode fall velocity reaches 2.81 m s$^{-1}$.

These transfer functions were assessed using accumulation measurements from an unshielded precipitation gauge and DFAR
gauge over 30-minute precipitation events during two winter seasons at the CARE test site in Egbert, ON, Canada. Estimates of fall velocity were provided by the POSS upward-facing Doppler radar.

All transfer functions presented in this study improved the agreement between the 30-minute adjusted precipitation accumulation values and DFAR reference values relative to the $K_{Universal}$ and $K_{CARE}$ transfer function based on mean wind speed and air temperature. The CFD transfer function agreed well with experimental results over all observed fall velocities
supporting the use of the modelling approach in Part I. The HE1 transfer function captured the collection efficiency trends for rain and snow well, with the collection efficiency for rain close to 1 and the collection efficiency for snow decreasing with wind speed. The HE2 transfer function better captured the collection efficiency for mixed precipitation with fall velocities between 1.2 m s$^{-1}$ to 4.6 m s$^{-1}$. Site-specific transfer functions ($K_{CARE}$) based on wind speed and temperature can also be employed to reduce the RMSE of measurements from unshielded weighing gauges relative to universal functions. The most
significant reductions in RMSE, however, were observed for the transfer functions based on wind speed and hydrometeor fall velocity.

The results of this study further demonstrate the important role of fall velocity on collection efficiency shown in previous studies (Nešpor and Sevruk, 1999;Thériault et al., 2012). Adjustment approaches incorporating fall velocity show tremendous value and potential, particularly in the general situation where DFAR measurements are not feasible, and can be applied where
the precipitation type is complex (e.g. snow transitioning to rain), uncertain, or even unknown. These approaches warrant further investigation at different sites with different precipitation characteristics, fall velocities, and wind speeds. Further study

to assess the collection efficiency relationships with wind speed and precipitation fall velocity for different shield configurations, as well as assessing the fall velocity using other means, including disdrometers or remote sensing, is also recommended.


*Disclaimer.* Many of the results presented in this work were obtained as part of the Solid Precipitation Intercomparison Experiment (SPICE) conducted on behalf of the World Meteorological Organization (WMO) Commission for Instruments and Methods of Observation (CIMO). The POSS was not included as part of the SPICE intercomparison. The analysis and views described herein are those of the authors, and do not represent the official outcome of WMO-SPICE. Mention of commercial companies or products is solely for the purposes of information and assessment within the scope of the present work, and does not constitute a commercial endorsement of any instrument or instrument manufacturer by the authors or the WMO.

*Author contribution.* J.H. was the lead author and was responsible for the methodology, analysis, visualization, and manuscript preparation and editing. M.E.E. provided guidance for the methodology, analysis, visualization, and writing – review and editing. P.I.J. provided guidance for the analysis, interpretation of results, visualization, and writing – review and editing.

*Acknowledgements.* The authors would like to acknowledge the encouragement and support of Rodica Nitu for this field of study. Thank-you to Christine Best, Pierrette Blanchard, and Sorin Pinzariu for supporting this work and Brian Sheppard for helpful discussions regarding the POSS. Thank-you to Hagop Mouradian, Sorin Pinzariu, and Lillian Yao for the data logger programming, electrical wiring, site maintenance, data ingest, and quality control for the CARE test site. The authors would also like to thank the WMO-SPICE team for their contributions and for discussions inspiring many facets of this work.

*Data availability.* The unshielded and reference event accumulations, wind speed, temperature, mean and mode fall velocity, and precipitation type data used in this study will be made available in a suitable online repository.


*Competing interests.* The authors declare that they have no conflict of interest.

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
