# Peer review of "Unshielded Precipitation Gauge Collection Efficiency with Wind Speed and Hydrometeor Fall Velocity. Part II: Experimental Results"

_Hydrology and Earth System Sciences, 2020_

## Referee Comment (RC1) · Anonymous Referee #1 · 4 Jan 2021

This manuscript shows that the RMSE of the collection efficiency can be significantly reduced if the fall speed derived from the Precipitation Occurrence Sensor System (POSS) is used. The paper is well written and shows new findings as the POSS can be used to improve the adjustment of solid precipitation. Nevertheless, I think that the text could be more concise for clarity and key information are missing. They are listed below. I recommend major revisions.

Major comments:

1. Introduction:

i) A few references are missing. 1) Colli et al. (2020) should be added to the paragraph

discussing methods to improve the adjustment of solid precipitation. Colli et al. (2020) showed that the precipitation intensity improvements the adjustment of solid precipitation at given wind speed. 2) Chubb et al. (2015) also proposed that the precipitation rate as could be used to adjust solid precipitation measurements.

Colli, M., Stagnaro, M., Lanza, L. G., Rasmussen, R. and Thériault, J. M. (2020). Adjustments for wind-induced undercatch in snowfall measurements based on precipitation intensity, Journal of hydrometeorology, 21, 1039-1050.

Chubb, T., Manton, M. J., Siems, S. T., Peace, A. D., & Bilish, S. P. (2015). Estimation of Wind-Induced Losses from a Precipitation Gauge Network in the Australian Snowy Mountains, Journal of Hydrometeorology, 16(6), 2619-2638.

ii) What is the goal of the study? A summary of the methodology is given in the last few paragraphs but it never stated the goal clearly.

2. The methodology section is incomplete. i) a description of the CFD simulations is missing. The relevant information from Part 1 should be added to the methodology of this manuscript. ii) A description of the method used to develop the transfer functions, in particular, the fall speed threshold values given in Section 3.1 should be added.

3. Section 3.1: How are the air temperature and fall speed threshold values determined in the study? In Table 2, the fall speed values for the precipitation type categories overlap. For example, snow events could also be mixed events if the temperature is < 0.5°C and the precipitation falls at < 2.32 m/s. It should be clarified in the text.

4. Why not using the temperature thresholds used in Kochendorfer et al. 2017b, which are -2°C to +2°C, to discriminate the precipitation types? Those are the threshold commonly used in the literature.

Minor comments:

1. Lines 81-83: Change hydrometeor type for "type of solid precipitation" or "type of snow" because the study was done for solid precipitation. Add "fall speed" to the

sentence because that is a key parameter of the study. The revised sentence could be: "Theriault et al. (2012) demonstrated similar trends for snowfall, with collection efficiencies varying significantly with the type of solid precipitation, fall speed and size distribution."

2. Lines 171-173: The transfer function uses the accumulated precipitation while the CFD simulations uses the precipitation intensity. Clarify this possible inconsistency.

3. Equation 3: Could you explain why this equation is relevant? If not, remove it.

4. Lines 287-292: Why using 1.93 m/s as a threshold? It should be explained.

5. Lines 296-301: Why using 2.81 m/s as a threshold? It should be explained.

6. Figure 4: Did you try using boxplots instead of a scatter plot to show the collection efficiency? It could give an idea of the scatter in the collection efficiency with wind speed.

7. Tables 3 to 9 could be put in an Appendix since that it is showing additional information. One could also do barplots instead of Tables.

8. Lines 477-479: The sentence: "While automatic . . . this work" seemed out of place. It may be better in the conclusion?

9. Line 505: The sentence: "The HE1 transfer function showed good results for snow, supporting its use for unshielded gauge.". I agree but Figure 3b (as an example) still shows lots of scatter in the collection efficiency for fall speeds associated with snow/solid precipitation ($\sim$1-2 m/s). Add a short discussion?

10. Lines 537-539: This sentence is not quite right and I think that it is an important point. The references from Kienzle (2008) and Harder and Pomeroy (2013) should be after the word "instructive" because they developed a method to diagnose the precipitation phase at the surface when the information aloft is not available. Theriault et al. (2012) suggested to use surface temperature but did not develop a method to diagnose

the type/phase of precipitation. At the end of the sentence, the authors should refer to a paper that state the importance of the atmospheric conditions aloft to determine the type/phase of precipitation at the surface such as for example Stewart et al. (2015).

Stewart, R. E., J. M. Theriault, and W. Henson, 2015: On the characteristics of and processes producing winter precipitation types near 0°C. Bull. Amer. Meteor. Soc., 96, 623–639, doi:10.1175/BAMS-D-14-00032.1.

---

## Referee Comment (RC2) · John Kochendorfer (Referee) · 13 Jan 2021

**General comments**

Part II of, "Unshielded precipitation gauge collection efficiency with wind speed and hydrometeor fall velocity" is the experimental companion to the Part I paper, which describes a modelling experiment. Part II tests the transfer function created in Part I, and it goes further to modify this transfer function based on the experimental results. It demonstrates that hydrometeor fall velocity can be used in a practical way to improve the adjustment of unshielded precipitation measurements. These improvements are impressive and significant.

Like Part I, the manuscript is well-written and easy to follow, and it is definitely worth publishing.

**Specific comments**

Ln. 65 – 67. This is a misinterpretation of those results. In addition to the uncertainty of the adjustment, it overlooks the fact that adjusted measurements increase the magnitude of errors multiplicatively. For example, if the gauge measurement has an inherent uncertainty of 0.1 mm, with CE = 0.5, after adjustment the uncertainty will be doubled along with the measurement. Two single Alter gauges agreeing with each other with an uncertainty of 0.09 mm does not imply that they can be adjusted without increasing the uncertainty. I accept that there is significant room for improvement in our transfer functions, but I find it very difficult to believe that adjusted unshielded measurements will ever be as accurate as well-shielded measurements. I am afraid that someone reading between the lines here might take that to be the suggestion.

Ln. 112. Change, "using similar methodology" to, "using *a* similar methodology" or, "using similar *methods*."

Ln. 172 and Eq (2). Why was $h$ chosen for precipitation, instead of $P$?

Ln. 269 – 270. This makes me wonder about the details and physics of the POSS averaging. How is the hydrometeor fall velocity calculated by the POSS when there is mixed precipitation, and/or when there is significant variability in the types of hydrometeors simultaneously present? I am guessing that for the purposes of transfer functions, ideally the fall velocity would be representative of the total mass of water falling, but perhaps it is actually weighted towards the average by volume?

Ln. 289. I apologize in advance, because I hate it when reviewers ask me these types of questions, but how was the threshold fall velocity of 1.93 m s$^{-1}$ selected?

Equation 7b. Given my comments on Part I this should come as no surprise, but I think that defining CE = 0.0 any under conditions is problematic.

Ln. 299. Clarify that $CE_{HE2}$ decreases linearly with wind speed *at a given/fixed hydrometeor fall velocity*.

Ln. 299 – 300. Explain how this works in practice. How were measurements that occurred when fall velocity was defined as zero treated? Were they simply removed from the analysis? How is the user of these functions supposed to adjust such measurements?

Ln. 314 – 315, Figure 4 caption. Typo. I believe that the three occurrences of "$u_p$" in, "fall velocity $u_p$ categories…" should be replaced with "$u_f$".

Ln. 352. Why wasn't the same temperature threshold technique used for $K_{Universal}$? At the risk of personifying a, "get off my lawn" attitude, I wonder how much of the improved performance of the $K_{CARE}$ adjusted measurements were caused by large errors in measurements that were over-adjusted using $K_{Universal}$ above this temperature threshold? The largest improvement in RMSE includes some of these measurements, when T is between positive and negative 2 deg C (Table 8), and I am guessing that at least some of the very poorly measurements were warmer, larger events (Fig. 5b).

Ln. 504. A realistic vertical wind profile, with a zero-slip boundary condition at the Earth's surface, may be important for larger wind shields.

Ln. 507 – 509. I agree that it is difficult to accurately adjust measurements at windy sites, but the 'limitation' described here is entirely avoidable. The collection efficiency was defined as zero above 7.19 m s$^{-1}$ by choice, not by necessity.

---

## Author Comment (AC1) · 1 Mar 2021

**Unshielded precipitation gauge collection efficiency with wind speed and hydrometeor fall velocity. Part II: experimental results**

**Author Response to Anonymous Referee #1**

This manuscript shows that the RMSE of the collection efficiency can be significantly reduced if the fall speed derived from the Precipitation Occurrence Sensor System (POSS) is used. The paper is well written and shows new findings as the POSS can be used to improve the adjustment of solid precipitation. Nevertheless, I think that the text could be more concise for clarity and key information are missing. They are listed below. I recommend major revisions.

Major comments:

1. Introduction:

i) A few references are missing. 1) Colli et al. (2020) should be added to the paragraph discussing methods to improve the adjustment of solid precipitation. Colli et al. (2020) showed that the precipitation intensity improvements the adjustment of solid precipitation at given wind speed. 2) Chubb et al. (2015) also proposed that the precipitation rate as could be used to adjust solid precipitation measurements.

Colli, M., Stagnaro, M., Lanza, L. G., Rasmussen, R. and Thériault, J. M. (2020). Adjustments for wind-induced undercatch in snowfall measurements based on precipitation intensity, Journal of hydrometeorology, 21, 1039-1050.

Chubb, T., Manton, M. J., Siems, S. T., Peace, A. D., & Bilish, S. P. (2015). Estimation of Wind-Induced Losses from a Precipitation Gauge Network in the Australian Snowy Mountains, Journal of Hydrometeorology, 16(6), 2619-2638.

**Authors' response:** We thank the reviewer for identifying these references, and will add them to the introduction.

ii) What is the goal of the study? A summary of the methodology is given in the last few paragraphs but it never stated the goal clearly.

**Authors' response:** We will state the goal of the study more clearly in the introduction: "In this work, transfer functions incorporating hydrometeor fall velocity are developed to reduce the uncertainty (RMSE) in collection efficiency and precipitation accumulation estimates from unshielded Geonor T-200B3 precipitation gauges." The authors also propose stating the goal earlier in the introduction, instead of only in the last paragraph.

2. The methodology section is incomplete.

i) a description of the CFD simulations is missing. The relevant information from Part 1 should be added to the methodology of this manuscript.

**Authors' response:** We recommend that a brief description of the CFD model and simulations is added to the methodology introducing the CFD transfer function (Sect. 2.4). We are wary of too much overlap with the Part I manuscript, which includes a detailed description of the CFD model and simulations. Within the present manuscript (Part II), the CFD transfer function is presented in the introduction (ln. 96-101) and methodology (ln. 208-216), with reference to the Part I manuscript.

ii) A description of the method used to develop the transfer functions, in particular, the fall speed threshold values given in

Section 3.1 should be added.

**Authors' response:** We will clarify this in the manuscript. The fall velocity and temperature ranges presented by precipitation phase in Section 3.1 (Table 2) summarize the event-based experimental observations from the POSS and a temperature sensor in an aspirated shield, respectively, and are independent from the methodology used to develop the transfer functions. The descriptions of the methods used to develop the HE1 and HE2 transfer functions in Section 3.3 should be expanded to include more detail regarding the fall velocity threshold values. For the HE1 function, the fall velocity threshold was varied over the measured fall velocity range in 0.01 m s$^{-1}$ increments, with the threshold of 1.93 m s$^{-1}$ found to provide the lowest overall

RMSE for the experimental dataset. For the HE2 transfer function, the fall velocity threshold was varied over the measurement fall velocity range in 0.01 m s$^{-1}$ increments, with the threshold of 2.81 m s$^{-1}$ found to provide the lowest overall RMSE. Details regarding the wind speed threshold for the CFD transfer function are provided in the Part I manuscript (Sect. 3.3), but can be reiterated in Section 2.4 of the present manuscript for clarity. For the KCARE transfer function, ln. 202-205 in the manuscript describes the methodology for determining the temperature threshold $T_t$.

3. Section 3.1: How are the air temperature and fall speed threshold values determined in the study?

**Authors' response:** The derivation of the air temperature and fall velocity thresholds used in the study are addressed in the response to comment 2ii above.

In Table 2, the fall speed values for the precipitation type categories overlap. For example, snow events could also be mixed events if the temperature is <0.5_C and the precipitation falls at < 2.32 m/s. It should be clarified in the text.

**Authors' response:** We agree to clarify this in the text. In Table 2, the temperature and fall velocity values are stratified by the 30-minute precipitation type classification determined from the minutely POSS precipitation type output following the methodology outlined in Sect. 2.3. As noted in the above response (comment 2ii), the experimental results summarized in Table 2 and plotted in Figure 1 are not used to determine threshold values for transfer functions. These results are presented in Section 3.1 to illustrate how multiple precipitation types, with different fall velocities, can be present within a given temperature range, presenting a challenge for transfer function methods distinguishing different precipitation types by temperature. The fall velocity thresholds for HE1 and HE2 were determined empirically to best capture the trends in experimental results by minimizing the RMSE.

4. Why not using the temperature thresholds used in Kochendorfer et al. 2017b, which are -2_C to +2_C, to discriminate the precipitation types? Those are the threshold commonly used in the literature.

**Authors' response:** The results in this study illustrate the challenges of using ambient temperature as a proxy for precipitation type, as multiple precipitation types – with different fall velocities – can be present within a given temperature range. Precipitation types and fall velocities in this study were determined from the POSS instrument as described in Sect. 2.3. Fig. 1 shows the event-based results with 30-minute mean surface air temperature and fall velocity by POSS precipitation type classification. It is apparent that in this -2 °C to +2 °C temperature range, a wide range of fall velocities and precipitation types can be present. Accordingly, there is significant scatter in the collection efficiency results with respect to wind speed for this temperature range, as shown in Fig. 2c.

The results in Tables 5 and 7 demonstrate that collection efficiencies and adjusted precipitation accumulation can be determined with greater certainty (lower RMSE) at these temperatures using adjustments based on wind speed and fall velocity relative to adjustments based on wind speed and temperature. The use of fall velocity provides a quantitative means for adjustments to be performed across precipitation types (for example, mixed precipitation with a range of fall velocities) and enables adjustments to be performed even under conditions where the precipitation type may be unknown or difficult to determine (e.g. 'undefined' events).

Minor comments:

1. Lines 81-83: Change hydrometeor type for "type of solid precipitation" or "type of snow" because the study was done for solid precipitation. Add "fall speed" to the sentence because that is a key parameter of the study. The revised sentence could be: "Theriault et al. (2012) demonstrated similar trends for snowfall, with collection efficiencies varying significantly with the type of solid precipitation, fall speed and size distribution."

**Authors' response:** We apologize for any confusion – this statement was made within the context of previous work involving CFD simulations. The simulations presented in Theriault et al. (2012) investigated how collection efficiency varies with wind speed depending on the specific snowflake type and selected slope size distribution value. Here, we can change "hydrometeor type" to "type of solid precipitation," as proposed. The linkage of the simulation results to theoretical terminal velocities computed for snowflakes that were collected and photographed is captured in lines 82 to 84 of the present manuscript.

2. Lines 171-173: The transfer function uses the accumulated precipitation while the CFD simulations uses the precipitation intensity. Clarify this possible inconsistency.

**Authors' response:** The CFD simulations are based on time-averaged simulation results and the collection efficiency is derived from the ratio of the precipitation intensity captured by the gauge to the true precipitation intensity falling in air.

Integrating over a period of time (in this case 30-minutes) gives the collection efficiency as a function of the ratio of the precipitation accumulation captured by the gauge to the true amount.

3. Equation 3: Could you explain why this equation is relevant? If not, remove it.

**Authors' response:** Equation 3 shows how the uncertainty in the experimental collection estimate scales with the magnitude of precipitation accumulation for rain, as shown in Fig. 2a and discussed in Section 3.2. It is apparent from Eq. 3 and the results in Fig. 2a that as the measured precipitation accumulations become smaller and approach the precipitation gauge measurement uncertainty, the uncertainty in the measured collection efficiency estimates can become quite large. This is an important point for understanding a component of the scatter in the collection efficiency results in Figs. 2b, 2c, and 2d, which is not readily apparent when collection efficiency results are plotted as a function of wind speed.

4. Lines 287-292: Why using 1.93 m/s as a threshold? It should be explained.

**Authors' response:** We will update Sect. 3.3 with this explanation. The threshold of 1.93 m s$^{-1}$ was determined by varying the fall velocity threshold in 0.01 m s$^{-1}$ increments over the measurement range of fall velocities (Table 2). This mean fall velocity threshold provided the lowest RMSE for the HE1 transfer function.

5. Lines 296-301: Why using 2.81 m/s as a threshold? It should be explained.

**Authors' response:** We will update Sect. 3.3 with this explanation. The threshold of 2.81 m s$^{-1}$ was determined by varying the fall velocity threshold in 0.01 m s$^{-1}$ increments over the measurement range of fall velocities (Table 2). This mean fall velocity threshold provided the lowest RMSE for the HE2 transfer function.

6. Figure 4: Did you try using boxplots instead of a scatter plot to show the collection efficiency? It could give an idea of the
scatter in the collection efficiency with wind speed.

**Authors' response:** Yes, this approach was considered. While the use of boxplots is useful for summarizing the distribution
of collection efficiencies across wind speed classes, or even wind speed and other classifications, it makes it more difficult to
trace the results for specific events across different classifications (e.g. precipitation type, temperature, and fall velocity)
because the events become lumped into boxes with only outliers shown. For example, looking at Fig. 2a, the two collection
efficiencies for rain above 1.3 correspond with very small accumulation values as discussed earlier (i.e. their values approach
the gauge measurement uncertainty). Looking at Fig. 2b, these events occur near 2 m s$^{-1}$ and 5 m s$^{-1}$. Fig. 2c shows that one of
these events is between -2 °C to 2 °C and one event is above 2 °C. Fig. 2d shows that both of these events have fall velocities
above 2.5 m s$^{-1}$. The RMSE values summarized in Tables 3, 5, 6, 8, and 9 provide a useful measure of the scatter, as they
capture the spread/scatter between the measurement and transfer function as the transfer functions change continuously with
wind speed and temperature or fall velocity.

7. Tables 3 to 9 could be put in an Appendix since that it is showing additional information. One could also do barplots instead
of Tables.

**Authors' response:** The authors appreciate the suggestion, but strongly recommend that Tables 3 to 6 remain in results Sect.
3.4 (Assessment of transfer functions: collection efficiency) and Tables 7 to 9 remain in results Sect. 3.5 (Assessment of
transfer functions: precipitation accumulation). The results in Table 3 capture the overall transfer function results and
demonstrate the improvement in the fall velocity transfer functions relative to current adjustments based on wind speed and
temperature. The other Tables demonstrate collection efficiency and precipitation accumulation RMSE by precipitation type,
temperature and fall velocity classifications, linking with the results and discussion associated with Figs. 4 and 5. The use of
Tables instead of bar plots has the advantage that the specific RMSE values are clearly shown for comparison with future
studies.

8. Lines 477-479: The sentence: "While automatic . . . this work" seemed out of place. It may be better in the conclusion?

**Authors' response:** We feel that this statement fits best within the context of the Discussion, where it follows the discussion
of the time periods and accumulation thresholds used in this and other work, and establishes boundaries for the scope of this
work. We agree that it could also work well in the Conclusions section, but it would be more challenging to establish the same
context in that case.

9. Line 505: The sentence: "The HE1 transfer function showed good results for snow, supporting its use for unshielded gauge.".

I agree but Figure 3b (as an example) still shows lots of scatter in the collection efficiency for fall speeds associated with snow/solid precipitation (_1-2 m/s). Add a short discussion?

**Authors' response:** This is a good point, and one that we believe is already discussed in the manuscript. Based on the 0.10

collection efficiency RMSE for snow events as identified by the POSS in Table 4, the HE1 transfer function showed good results, as stated in line 505. Looking at the 0.10 collection efficiency RMSE for HE1 at fall velocity values ≤ 1.5 m/s in Table

6 tells a similar story. However, in line with the reviewer's point, the collection efficiency RMSE for HE1 in Table 6 is higher (0.15) for events with fall velocity values between 1.5 m/s and 2 m/s. This higher RMSE value for HE1 is consistent with that for events classified as mixed precipitation in Table 4. This limitation of HE1 is noted and discussed in lines 516-521 of the manuscript.

10. Lines 537-539: This sentence is not quite right and I think that it is an important point. The references from Kienzle (2008)

and Harder and Pomeroy (2013) should be after the word "instructive" because they developed a method to diagnose the precipitation phase at the surface when the information aloft is not available. Theriault et al. (2012) suggested to use surface temperature but did not develop a method to diagnose the type/phase of precipitation. At the end of the sentence, the authors should refer to a paper that state the importance of the atmospheric conditions aloft to determine the type/phase of precipitation at the surface such as for example Stewart et al. (2015).

Stewart, R. E., J. M. Theriault, and W. Henson, 2015: On the characteristics of and processes producing winter precipitation types near 0_C. Bull. Amer. Meteor. Soc., 96, 623–639, doi:10.1175/BAMS-D-14-00032.1.

**Authors' response:** Thank-you for pointing this out. We will update the references as suggested to improve the clarity of this sentence.

---

## Author Comment (AC2) · 1 Mar 2021

**Unshielded precipitation gauge collection efficiency with wind speed and hydrometeor fall velocity. Part II: experimental results**

**Author Response to J. Kochendorfer (Referee #2)**

General comments

Part II of, "Unshielded precipitation gauge collection efficiency with wind speed and hydrometeor fall velocity" is the experimental companion to the Part I paper, which describes a modelling experiment. Part II tests the transfer function created in Part I, and it goes further to modify this transfer function based on the experimental results. It demonstrates that hydrometeor fall velocity can be used in a practical way to improve the adjustment of unshielded precipitation measurements. These improvements are impressive and significant.

Like Part I, the manuscript is well-written and easy to follow, and it is definitely worth publishing.

**Authors' response:** Thank-you!

Specific comments

Ln. 65 – 67. This is a misinterpretation of those results. In addition to the uncertainty of the adjustment, it overlooks the fact that adjusted measurements increase the magnitude of errors multiplicatively. For example, if the gauge measurement has an inherent uncertainty of 0.1 mm, with CE = 0.5, after adjustment the uncertainty will be doubled along with the measurement. Two single Alter gauges agreeing with each other with an uncertainty of 0.09 mm does not imply that they can be adjusted without increasing the uncertainty. I accept that there is significant room for improvement in our transfer functions, but I find it very difficult to believe that adjusted unshielded measurements will ever be as accurate as well-shielded measurements. I am afraid that someone reading between the lines here might take that to be the suggestion.

**Authors' response:** Dr. Kochendorfer makes a good point here. We will remove the reference to the comparison of replicate configurations of weighing gauges (Ln. 65-67).

Ln. 112. Change, "using similar methodology" to, "using *a* similar methodology" or, "using similar *methods*."

**Authors' response:** Updated to "using a similar methodology".

Ln. 172 and Eq (2). Why was *h* chosen for precipitation, instead of *P*?

**Authors' response:** *h* was originally chosen to refer to precipitation accumulation as a height in units of mm. *h* has been revised to *P* to make the linkage with precipitation clearer and to match the terminology of previous publications. Thank-you.

Ln. 269 – 270. This makes me wonder about the details and physics of the POSS averaging. How is the hydrometeor fall velocity calculated by the POSS when there is mixed precipitation, and/or when there is significant variability in the types of hydrometeors simultaneously present? I am guessing that for the purposes of transfer functions, ideally the fall velocity would be representative of the total mass of water falling, but perhaps it is actually weighted towards the average by volume?

**Authors' response:** The POSS is an X Band (3cm wavelength) radar that measures the Doppler velocity spectrum from which the hydrometeor size distribution is derived. This has been described in detail in previous publications, including its use for precipitation typing; we refer the reviewer to the following publications for the details (Sheppard, 1990; Sheppard and Joe,

1994, 2000, 2008). The advantage of the POSS is that it rapidly measures the Doppler spectrum from a very large volume compared to other disdrometers, which measure individual particles with more limited sampling (e.g. Thies LPM, OTT

Parsivel2). For large hydrometeors (say 5 mm), the sample volume is about the size of a small room. Several hundred

Doppler/hydrometeor spectra are measured and reported every minute. There is on-ongoing research for snow and mixed precipitation type retrievals. We agree that ideally, the fall velocity would be representative of the total mass of water falling, but the complexities of hydrometeor drag, density, and mass are confounding factors still to be resolved. While the present approach of estimating the event fall velocity from the 30-minute average appears to perform well overall, further study to better characterize the fall velocity distribution and changes over 30-minute time periods could lead to further improvements in the model under specific conditions such as mixed precipitation.

Ln. 289. I apologize in advance, because I hate it when reviewers ask me these types of questions, but how was the threshold fall velocity of 1.93 m s$^{-1}$ selected?

**Authors' response:** The threshold of 1.93 m s$^{-1}$ was determined by varying the fall velocity threshold in 0.01 m s$^{-1}$ increments over the measurement fall velocity range in Table 2. This mean fall velocity threshold provided the lowest RMSE for the HE1

transfer function. A similar approach was used to derive the fall velocity threshold for HE2. We will add this information to the manuscript.

Equation 7b. Given my comments on Part I this should come as no surprise, but I think that defining CE = 0.0 any under conditions is problematic.

**Authors' response:** Dr. Kochendorfer raises an important issue with the definition of the collection efficiency at high wind speeds in the transfer function. The authors recommend revising Eq. 7b, Table 1, and Fig. 4c for HE1 with a minimum collection efficiency of 0.2 and wind speed threshold of 5.75 m s$^{-1}$, following the general approach of Kochendorfer et al.

(2017).

Ln. 299. Clarify that *CEHE2* decreases linearly with wind speed *at a given/fixed hydrometeor fall velocity*.

**Authors' response:** Updated. Thank-you.

Ln. 299 – 300. Explain how this works in practice. How were measurements that occurred when fall velocity was defined as zero treated? Were they simply removed from the analysis? How is the user of these functions supposed to adjust such measurements?

**Authors' response:** Over the test period there were no fall velocities of zero reported by the POSS and 30-minute mean fall velocities were ~1 m s$^{-1}$ or higher. During non-precipitating periods the POSS does not output a fall velocity and these periods are not included in the 30-minute average. While fall velocities of zero were not encountered during this study, and would not be expected in general, the *HE2* transfer function is still defined in this case. In the case of zero fall velocity the collection efficiency decreases with wind speed alone as shown in Eq. 8a. In this case the collection efficiency decrease with wind speed will be faster than that for conditions where the fall velocity is higher.

Ln. 314 – 315, Figure 4 caption. Typo. I believe that the three occurrences of "*up*" in,"fall velocity *up* categories…" should be replaced with "*uf*".

**Authors' response:** Updated. Thank-you.

Ln. 352. Why wasn't the same temperature threshold technique used for *KUniversal*? At the risk of personifying a, "get off my lawn" attitude, I wonder how much of the improved performance of the *KCARE* adjusted measurements were caused by large errors in measurements that were over-adjusted using *KUniversal* above this temperature threshold? The largest improvement in RMSE includes some of these measurements, when T is between positive and negative 2 deg C (Table 8), and

I am guessing that at least some of the very poorly measurements were warmer, larger events (Fig. 5b).

**Authors' response:** *KUniversal* was developed from the WMO-SPICE results for eight test sites and is used for comparison with the present study results from the CARE field test site. Modifications to *KUniversal* using the temperature threshold technique would need to be assessed based on the entire dataset (all eight sites) and is beyond the scope of this study. *KCARE*

is developed from the CARE dataset for comparison with the site-specific fall velocity transfer functions developed in this study. Both *KUniversal* and *KCARE* are similar at colder temperatures but differ as the temperature increases. The improvement in the *KCARE* transfer function results are primarily attributed to this more rapid increase in collection efficiency with temperature, reducing the overadjustment of some events and increasing the underadjustment of some events between -5

°C and -2 °C and between -2 °C and 2 °C (as shown in Fig. 5 and Table 8). It is important to note that even the *KCARE* transfer function exhibits increased uncertainties at these warmer temperatures relative to transfer functions using fall velocity, as rain, mixed precipitation, and snow can occur with different collection efficiencies. These differences cannot be distinguished using temperature alone, resulting in increased uncertainties at these temperatures.

Ln. 504. A realistic vertical wind profile, with a zero-slip boundary condition at the Earth's surface, may be important for larger wind shields.

**Authors' response:** Thank-you. This is an important point for studying other shield and gauge combinations in the future.

This note will be added to the manuscript.

Ln. 507 – 509. I agree that it is difficult to accurately adjust measurements at windy sites, but the 'limitation' described here is entirely avoidable. The collection efficiency was defined as zero above 7.19 m s-1 by choice, not by necessity.

**Authors' response:** We will revise the discussion for the HE1 transfer function to include a transfer function minimum collection efficiency of 0.2 for wind speeds above 5.75 m s$^{-1}$ following the general approach of Kochendorfer et al. (2017).